# Smart Hydrogels: Preparation, Characterization, and Determination of Transition Points of Crosslinked *N*-Isopropyl Acrylamide/Acrylamide/Carboxylic Acids Polymers

**DOI:** 10.3390/gels7030113

**Published:** 2021-08-08

**Authors:** Yasemin Işıkver, Dursun Saraydın

**Affiliations:** Chemistry Department, Science Faculty, Sivas Cumhuriyet University, Sivas 58140, Turkey; caldiran@cumhuriyet.edu.tr

**Keywords:** smart hydrogel, NIPAAm, LCST, transition point, sigmoidal equation, human serum albumin

## Abstract

Smart hydrogels (SH) were prepared by thermal free radical polymerization of *N*-isopropyl acrylamide (NIPAAm), acrylamide (AAm) with acrylic acid (A) or maleic acid (M), and *N*,*N*′-methylene bisacrylamide. Spectroscopic and thermal characterizations of SHs were performed using FTIR, TGA, and DSC. To determine the effects of SHs on swelling characteristics, swelling studies were performed in different solvents, solutions, temperatures, pHs, and ionic strengths. In addition, cycle equilibrium swelling studies were carried out at different temperatures and pHs. The temperature and pH transition points of SHs are calculated using a sigmoidal equation. The pH transition points were calculated as 5.2 and 4.2 for SH-M and SH-A, respectively. The NIPAAm/AAm hydrogel exhibits a critical solution temperature (LCST) of 28.35 °C, while the SH-A and SH-M hydrogels exhibit the LCST of 34.215 °C and 28.798 °C, respectively, and the LCST of SH-A is close to the body. temperature. Commercial (CHSA) and blood human serum albumin (BHSA) were used to find the adsorption properties of biopolymers on SHs. SH-M was the most efficient SH, adsorbing 49% of CHSA while absorbing 16% of BHSA. In conclusion, the sigmoidal equation or Gaussian approach can be a useful tool for chemists, chemical engineers, polymer and plastics scientists to find the transition points of smart hydrogels.

## 1. Introduction

Three-dimensional cross-linked polymeric structures that love water are called hydrogels [1,2,3], and those that respond to external stimuli (such as pH, temperature, ionic strength, electric field, light, magnetic field, and so forth) as volume changes are called smart or intelligent or stimuli-responsive hydrogels [4,5,6,7,8,9,10,11,12,13,14,15].

Temperature responsivity of smart hydrogels can be achieved with hydrophobic monomers ((such as *N*-isopropyl acrylamide, vinyl methylether, N-methyl acrylamide, *N*,*N*-dimethyl acrylamide, N-tert-butyl acrylamide), and pH and ionic responsivity can be achieved with ionizable organic acidic or basic monomers (such as methacrylic acid, crotonic acid, itaconic acid, mesaconic acid, aconitic acid or dimethyl amino ethyl methacrylate, diethyl amino ethyl methacrylate, vinyl pyrrolidone, vinyl imidazole) [4,5,6,7,8,9,10].

Smart hydrogels are used in the fields of environment, agriculture, biomedicine, and bioengineering, and they are also used as biomaterials since the transition temperature of temperature-sensitive polymers is close to body temperature. These gels are used in biological sensing, drug release and tissue degeneration, artificial muscles, micro-electro-mechanical-system instruments such as microvalves, and microfluidity controllers. In addition to these, diapers, irrigation, and smart windows can be given as examples in daily life [11,12,13,14,15].

In the temperature-responsive gels, when the temperature rises above a certain value, phase separation occurs, and the gel shrinks. This is due to the increased hydrophobic effect and the weakening of hydrogen bonds. This hydrophilic-hydrophobic transition temperature value is called the lowest critical solution temperature (LCST). Below the LCST, the compatibility of the gel with water increases, and swelling is observed. This is explained by the strengthening of hydrogen bonds between water molecules and hydrophilic groups [4,5,6,7,8,9,10,11,12,13,14,15].

The pH-sensitive gels containing acidic and basic groups show swelling or shrinkage behavior depending on pH. Groups that ionize by giving and receiving protons according to the pH of the environment form a constant electrical charge (negative or positive) in the gel. As the electrostatic forces formed repel each other, water entry into the gel structure becomes easier and swelling is observed. In a hydrogel carrying acidic groups, acidic groups do not ionize at low pH, but ionize at high pH and become negatively charged. The COO^−^ groups formed repel each other. Thus, water diffusion into the network takes place, and the hydrogel swells. In a hydrogel containing a basic group, it is protonated at low pH. The resulting positive charges repel each other, water diffusion is achieved, and swelling is observed in the gel. This swelling shrinkage transition pH is called the inflection point (IP) [4,5,6,7,8,9,10,11,12,13,14,15].

While the number of publications on hydrogels since 1975 is around 70,000, 6200 of these publications are on smart or stimuli-responsive hydrogels in SCI. These publication numbers show that the research and development of smart hydrogels are still in their infancy compared to the continued and significantly increased efforts to develop hydrogels [16]. However, despite the mention of LSCT in 812 of these publications, there are no studies on the determination of the transition points (i.e., such as LCST, IP) and the magnitude of the stimuli-responsivity, which are very important in the use of smart hydrogels as biomaterials, with appropriate equations.

To improve the swelling properties of N-isopropyl*N*-isopropyl acrylamide/acrylamide gels and to impart pH sensitivity it has been copolymerized for a monoprotic carboxylic acid (acrylic acid (A)), or a diprotic carboxylic acid (maleic acid (M)). The resulting gel is responsive to both temperature and pH changes. By copolymerization with a suitable monomer, the hydration degree and swelling kinetics of the final hydrogel are easily tuned. Acrylamide (AAm) is a monomer of the class poly(acrylamides), which upon polymerization gives a water-soluble polymer with no LCST. In general, acrylamide-based compounds are well known for their absorptive and medicinal properties. In line with our research interest in AAm based materials, [17,18,19,20,21,22] in this article we report the copolymerization of NIPAAm with AAm. The present study aims to develop *N*-isopropyl acrylamide/acrylamide; (NIPAAm/AAm) based hydrogels with pH and temperature sensitivities and higher swelling properties. To easily compare the effects of the stimulating parameters investigated, A and M were selected as acidic comonomers which were very similar in chemical structure with only one excess of the carboxyl group of unsaturated organic acids. The preparation and characterization of hydrogels, detailed analysis of their swelling behavior, determination of critical transition points using appropriate equations, and human serum albumin adsorption are presented in this report. A detailed understanding of the physical properties of these new ionizable hydrogels concerning molecular heterogeneity is an important criterion in targeted applications such as temperature-, pH-, ionic strength-, and solvent-responsive sorbent systems.

## 2. Results and Discussion

### 2.1. Preparation of Hydrogels

Hydrogels composed of NIPAAm, AAm, and carboxylic acid unit(s) containing comonomers (Table 1) were prepared by thermal free radical solution redox polymerization using *N*,*N*′-methylenebisacrylamide (NBis) as crosslinker. The polymerization mechanism is very well established and schematic representation is given in Figure 1. The polymerization and crosslinking mechanism took place simultaneously. Although gelation was occurring in about 1 h, the reaction let it proceed for 24 h, and finally, the hydrogel rods obtained in the plastic straws were cut into pieces of 3–4 mm length and washed with distilled water and dried in air and vacuum. Hydrogels were colorless and transparent. Hydrogels prepared from *N*-isopropyl acrylamide with acrylamide were labeled as SH-O, while gels containing acrylic acid were named SH-A, and gels containing maleic acid were named SH-M [23].

### 2.2. Characterization of SHs

#### 2.2.1. FTIR Analysis

In the FTIR spectra of the SHs (Figure 2), the typical absorption bands for NIPAAm, AAm, and A or M units can be seen about 3600–3100 cm^−1^ as broad bands for secondary NH amide, between 2980 and 2878 cm^−1^ for CH stretching frequencies for isopropyl groups, and at 1670 cm^−1^ a strong C=O amide I band, and at 1550 cm^−1^ for another strong amide II band. The peak around 1400–1300 cm^−1^ belongs to −CH(CH_3)2_ group in isopropylgroups [18,20,24]. From this spectral analysis, it can be concluded that a polymeric network is formed, because of the functional groups of each component of the hydrogel units: NIPAAm, AAm, and A or M monomers with NBis.

#### 2.2.2. Thermal Analysis

TG analysis

The thermal degradation values such as the initial degradation temperatures (T_i_), the temperature of maximum rate (T_max_), the degradation final temperature (T_f_), the half-life temperature (T_h_), and the maximum decomposition rate (r_max_), and the amount of the substance at the maximum rate (C_max_) values were found the thermograms of hydrogels (Figure 3) and were given Table 2.

All temperature values of SH-O were lower than the others. T_max_ and T_f_ values of SH-A were higher than those of SH-M. These values indicate that SH-M containing two carboxyl groups in its structural repeating unit degrades at slightly lower temperatures than SH-A containing a single carboxyl group. The presence of more than one carboxyl group in the structural repeating unit of SH slightly facilitated thermal degradation. r_max_ values range from 0.85 to 0.87 mg min^−1^, while C_max_ values range from 41 to 45%.

The Thermal Degradation Kinetics of the SHs

To determine the thermal decomposition of the samples, Freeman-Carroll (FC) and Jerez (J) methods were used [25].

Freeman-Carroll equation;
(1)$$Y=n-\frac{E}{R}X$$
can be derived, where
(2)$$X=\frac{\Delta \left(\frac{1}{T}\right)}{\Delta \ln\left(1-c\right)}\;Y=\frac{\Delta \ln\beta \left(\frac{\mathrm{dc}}{\mathrm{dT}}\right)}{\Delta \ln\left(1-c\right)}$$

In these equations, *n*; reaction order, E; activation energy of the degradation reaction, R; ideal gas constant, T; absolute temperature, β; heating rate and, c, mass fraction reacted, as symbolized.

In the Jerez method, the X and Y data of the Freeman-Carroll method are corrected with the mathematical form of an application (Figure 4). According to the Jerez method, the degradation kinetics parameters such as *n* and E are calculated for the following equations.
(3)$$n=\frac{\bar{Y}}{1-Q\bar{X}}\;\mathrm{and}\;E=\frac{\mathrm{RQ}\bar{Y}}{1-Q\bar{X}}$$
where $\bar{X}$ and $\bar{Y}$ are mean values X and Y, and Q is given as
(4)$$Q={\left[\frac{c^{\prime }T_{\max}^{2}}{1-c}\right]}_{c^{\prime }=\max}=\frac{E}{\mathrm{nR}}$$
here, $c^{\prime }=\frac{dc}{dT}=\frac{1}{\beta }\frac{dc}{dt}$ and DTG peak maximum transformation rate peak is calculated by reading.

Reaction order (*n*) and activation energy (E, kJ mol^−1^) values calculated via both FC and J methods are presented in Table 3.

While *n*_FC_ values are in a wide range of 0.53 to 4.18, *n*_J_ values are calculated as approximately zero for all smart hydrogels. The zero-order degradation kinetics is as suitable as possible for the decomposition of smart hydrogels since the degradation rate is independent of the number of unreacted solids.

Although the activation energies of smart hydrogels in both methods were highest in SH-O, it was observed that they shifted towards smaller energy values with the incorporation of acidic comonomers into the structure. The presence of a carboxyl group or groups in the structure reduces the minimum energy value required for the d decomposition of smart hydrogels. The E value of SH-A containing a monocarboxylic co-monomer is the lowest compared to the others. Therefore, it shows that SH-A is the most easily degradable hydrogel with an increase in temperature. In addition, the fact that the E value of SH-M is greater than that of SH-A can be considered as an increase in maleic acid containing-dicarboxylic groups in SH-M due to the repulsions between these groups.

In addition, the values found as a result of applying the Jerez correction method to the values found by the Freeman Carrol method are more realistic [18,20].

DSC analysis.

LCST and glass transition temperature (T_g_) of the SH were determined with the DSC thermograms (Figure 5) and the results are given in Table 4.

The LCST value of SH-O found from the first shoulders in the parts of DSC thermograms [26] between 15 and 45 °C (Figure 5) was found very close to 32 °C as expected [5]. However, while the LCST value of SH-A, which was found to be 36.58 °C, was around human body temperature, the LCST value of SH-M decreased to 27.41 °C. There may be a dramatic decrease in LCST due to repulsions between the double –COOH groups contained in maleic acid in SH-M.

As can be seen from Table 4, the T_g_ value of SH-O is 12–13 °C greater than the T_g_ values of both SH-A and SH-M. The –COOH group(s) in the structure of SHs act as plasticizers and cause the T_g_ value of both SH-A and SH-M to be lower than the T_g_ value of SH-O. Thus, carboxyl groups in SH reduce the T_g_ of SH-O.

#### 2.2.3. Swelling Experiments

The responsivities of smart hydrogels to stimuli such as pH, temperature, solvent type, solvent concentration, ionic strength was measured by measuring the fluid remaining in the hydrogel [20].

The equilibrium swelling; (S, g g_SH_^−1^) was calculated using the following relation:(5)$$S\%=\frac{a_{f}}{m_{o}}$$
where a_f_ is the mass of fluid absorbed by the gel (g), while m_o_ is the mass of the dry gel (g_SH_).

pH-responsive swelling

The polymeric networks containing ionizable functional groups exhibit pH responsivity [27]. The responsibility of surrounding media pH on the swelling values of SHs at 25 °C and 40 °C between pH = 2–9 with ionic strength of I = 0.05 M is shown in Figure 6.

The graphs in Figure 6 for SH-O gave a straight line, while the hydrogels containing carboxylic acid (SH-A and SH-M) gave sigmoidal curves. By adapting the sigmoidal 4 parameter relation [28,29,30,31] to determine the transition points (inflection point, IP) of pH responsive-swelling of hydrogels, the following equation can be written;
(6)$$S=S_{o\;}+\frac{\alpha }{1+e^{-\left(\frac{\mathrm{pH}-\mathrm{IP}}{\beta }\right)}\;}$$
where S_o_ (g g_SH_^−1^) and S_max_ (g g_SH_^−1^) are asymptotic lower and upper values of swelling (values that the function approaches but never quite reaches). The value α (g g_SH_^−1^) is the magnitude of the hydrogel’s responsivity, that is, the change in the swelling value of the stimulus and is equal to S_o_−S_max_. β is the slope parameter and IP is the inflection point of a hydrogel.

The pH-responsive swelling parameter of the SHs at 25 and 40 °C were calculated from Equation (6). The correlation coefficients of the graphs drawn according to the proposed equation are 0.996 and above, indicating that the sigmoidal 4-parameter equation can be used to find the transition points of smart hydrogels.

Table 5 presents the values of α, β, IP, and S_o_ with standard error (SE) found from Equation (6), as well as S_max_, IP_S_ (swelling in IP value), and experimentally found S_i_ and S_f_ (initial and swelling values).

Table 5 shows that the IP values of hydrogels containing carboxyl groups increase with increasing temperature and the IP value of SH-A is higher than that of SH-M. As expected, there was no major change in the pH-responsive swelling behavior of nonionic SH-O at temperatures above and below the LCST. Therefore, SH-O is not sensitive to pH. However, the swellings of SH-A and SH-M containing the carboxylic acid group(s) were largely dependent on the pH of the external environment. At low pH values, the degree of swelling is low as the carboxyl side group(s) in the main chains are not ionized and intermolecular complexation may occur via hydrogen bonds (physical cross-linking). As the degree of ionization increased above the nominal pK_a_ values of the carboxylic acid monomers for A (pK_a_ = 4.25), for M (pK_a1_ = 1.94 and pK_a2_ = 6.22), the increased hydrophilicity resulted in greater swelling values as reported for similar structures in the literature [18,24,27].

Temperature-responsive swelling behavior

To examine the temperature responsivity of smart hydrogels, swelling values were obtained at pH = 3 (below IP values) or pH = 8 (over IP values) and I = 0.05 M in the range of 10–60 °C, and temperature-swelling graphs were created, and are shown in Figure 7.

Hydrogels gave sigmoidal curves in temperature-responsive swelling [32] as well as in pH-responsive swelling. By adapting the sigmoidal 4 parameter relation [28,29,30,31] to determine the transition points (LCST, °C) temperature-responsive swelling of hydrogels, the following equation can be written;
(7)$$S=S_{o\;}+\frac{\phi }{1+e^{-\left(\frac{T-\mathrm{LCST}}{\Psi }\right)}\;}$$
where S_o_ (g g_SH_^−1^) and S_max_ (g g_SH_^−1^) are asymptotic lower and upper values of swelling (values that the function approaches but never quite reaches). The value ϕ (g g_SH_^−1^) is the magnitude of the hydrogel’s responsivity, that is, the change in the swelling value of the stimulus and is equal to S_o_−S_max_. Ψ is the slope parameter and LCST is the lower critical solution point of a hydrogel.

The temperature-responsive swelling parameters of the SHs at pH = 3 and 8 were calculated from Equation (7). The correlation coefficients of the graphs drawn according to the proposed equation are 1 or very close to one, indicating that the sigmoidal 4-parameter equation can be used to determine the transition points of smart hydrogels.

Table 6 presents the values of ϕ, Ψ, LCST, and S_o_ with standard error (SE) found from Equation (7), as well as S_max_, LCST_S_ (swelling in LCST value), and experimentally found S_i_ and S_f_ (initial and final swelling values).

The negative Ψ values of hydrogels at both pHs are due to the decrease in swelling with increasing temperature in the transition zone. This is because temperature-responsive SHs swell when cooled below the LCST and collapse when heated above the LCST [18,32]. While the LCST values at pH = 3 are approximately the same for all gels, they differ at pH = 8. This is because gels, especially SH-A, swell more at pH = 8 for the reasons mentioned earlier. The LCST value of SH-A approaches the body temperature, which is important for its potential to be used as a biomaterial. It is also seen that the LCST value found from the DSC thermogram of SH-A and the LSCT value found swelling are close to each other. The difference between the values found by two different methods may be due to the effect of the swelling medium.

In both pH values, the responsibility value (ϕ) of SH-O is lower than other hydrogels. The addition of acidic co-monomers to SH-O, which is not responsive to pH, increases swelling and increases the responsivity of SH-A and SH-M to temperature.

Ionic strength-responsive swelling behavior

The ionic strength plays an important role in the swelling behavior of hydrogels [33,34,35]. The ionic strength-responsive swelling behavior was investigated at various NaCl concentrations (mol L^−1^) at 25 °C and the ionic strength-responsive swelling of the hydrogels is shown in Figure 8.

As the ionic strength increased, the swelling value of SH-O did not change much, while the swelling values of SHs containing carboxylic acid group(s) decreased. This phenomenon can be attributed to the electrostatic repulsion between the charged groups in the network chain and the concentration difference between the mobile ions in the hydrogel and the external solution [27,36]. In addition, the swelling values of SH-A and SH-M fall below that of SH-O when they exceed a certain ionic strength value. When this limit value is exceeded, it may be that the cations in the environment prevent swelling by affecting the hydrogen bond formation as a result of the interaction of the carboxyl groups of the anionic hydrogels.

Hydrogels gave exponential decay curves in ionic strength-responsive swelling. By adapting the exponential decay; single, 3 parameter relation to determine the parameters of the ionic strength-responsive of hydrogels, the following equation can be written;
(8)$$S={\;S}_{o}+{\;\xi \;e}^{-\;\lambda \;I}$$
where I is ionic strength (mol L^−1^), and S is swelling (g g_SH_^−1^). S_o_ is offset swelling or minimum swelling value (g g_SH_^−1^), and ξ is swelling amplitude (g g_SH_^−1^) or the magnitude of the responsivity of smart hydrogels to ionic strength. λ is the exponential slope coefficient (mol^−1^ L) and is a measure of how the swelling decreases concerning ionic strength.

The ionic strength-responsive swelling parameters of the SHs were calculated from Equation (8). The correlation coefficients (r) of the graphs drawn according to the proposed equation were found in the range of 0.98 to 0.99. The r values close to the unity indicate that the exponential decay single, 3 parameters equation can be used to determine the ionic strength responsive swelling parameters of smart hydrogels.

In addition to the So, ξ and λ values with standard error (SE) found from the Equation (8), S_max_, and experimentally found S_i_ and S_f_ (initial and final swelling values) values are given in Table 7.

As seen in Table 7, the responsibility value (**ξ**) of SH-O is quite low compared to other hydrogels. Responsibility for ionic strength of SH-A (**ξ** = 1.92 g g_SH_^−1^) and SH-M (**ξ** = 1.14 g g_SH_^−1^) containing acidic co-monomers relative to SH-O (**ξ** = 0.15 g g_SH_^−1^), which is not sensitive to ionic strength has been more.

To compare the ion and counter ion influences on the swelling of SHs in solutions of NaNO_3_, NaCl, and CaCl_2_ with an ionic strength of 0.05 M, and in water, a bar graph is plotted and shown in Figure 9. The swelling ratios decreased according to the following sequence water, NaNO_3_, NaCl, CaCl_2_. As the SRH containing carboxylic acid group(s) were swelled in saline solutions, the –COOH groups were neutralized by the cations in the external solution, and the swelling ratios were decreased. When the fixed charges on polymeric side chains were fully neutralized, SHs containing carboxylic acid group(s) showed nonionic behavior like SH-O. In various saline solutions, hydrogels showed the Donnan effect when the charges on the polymeric side chain were neutralized and then showed a salting-out effect with the gels going to a nonionic state [34,35,36].

In the polymer chain, monovalent Na^+^ ions interact with a −COO^−^ unit, divalent Ca^2+^ ions interact with the two −COO^−^ units. NaCl solution and CaCl_2_ solution than in the polymeric chain results in the increased neutralization will reduce the loads on. Consequently, uncharged polymer chains act as polymer chains, and the swelling value decreases.

Also, according to Cl^−^ ions NO_3_^−^ ions stronger salting precipitant (salting out) due to the inducing action of water molecules around the polymer out of the degree of swelling of the polymer NaNO_3_ > NaCl was in the form.

Comparison of stimuli responsivities

To compare the responsibilities of SHs to stimuli such as temperature, pH, ionic strength, a bar chart was prepared and presented in Figure 10.

While all gels are temperature-responsible, SH-O is irresponsible to pH but has little ionic strength. Of the gels containing carboxyl groups, SH-A showed up to 1.8 to 3.1 times more responsibility for the temperature stimulus than pH, while SH-M showed up to 2.4 to 3.7 times more responsibility for the temperature stimulus. At the same time, SH-A and SH-M are 1.1–1.2 times and 2.0 times more responsible for temperature stimulus than Ionic strength stimulus, respectively. That is, the prepared gels are mostly responsible for the temperature stimulus, then the ionic strength stimulus, and finally the pH stimulus. The high NIPAAm ratio in hydrogels, the interaction of the carboxyl group(s) in acidic gels with the ions forming the ionic strength, and the low amount of acidic comonomers in the network may have resulted in an order of responsibility as T, I, and pH stimuli.

These hydrogels, which are prepared by using the temperature responsivity of *N*-isopropyl acrylamide monomer and the pH and ionic strength responsivity of carboxylic acid-containing monomers and the mechanical strength of acrylamide monomer, with a durable, homogeneous appearance, which are very sensitive to stimuli, can be called stimuli-responsive, intelligent or smart hydrogels.

Reversible swelling

Reversible swelling [37,38] experiments were performed for hydrogels at two extreme pH, below and above the dissociation values of carboxyl units, 3.0 and 8.0, and below and above the LCST values at two extreme temperatures, 25 and 40 °C. Equilibrium swelling values of each hydrogel in a buffered solution at different pHs for 24 h were determined and are presented in Figure 11. There is no change in swelling behavior in the reversible swelling of the neutral SH-O hydrogel at extreme pH and temperatures. Due to the carboxylic acid unit(s) in the SH-A and SH-M hydrogels, they exhibit pH-responsive swelling behavior. At pH = 8.0, the negatively charged hydrogel increases the electrostatic repulsion between the chains, and at pH = 3.0, the discharge/discharge reduces swelling. Therefore, hydrogels swell highly at high pH and relatively shrink at low pH of the medium. It was determined that the hydrogels retained their shape and integrity throughout the experiments (8 days). From this result, it can be assumed that the hydrogels are resistant to fragmentation.

Solvent concentration-responsive swelling

Swelling measurements were performed in dimethyl sulfoxide (DMSO)-water mixtures at 25 °C and are presented in Figure 12.

Hydrophobic isopropyl groups on the network chains are responsible for reentrant phase transitions of hydrogels in DMSO-water mixtures [39,40]. In DMSO-water mixtures, hydrophobic isopropyl groups in the hydrogel reduce the interaction between water and hydrogel, leading to a strong attraction of water molecules by DMSO. Thus, the water-DMSO interaction suppresses the water-hydrogel and DMSO-hydrogel interactions, causing the gels to shrink. When the DMSO concentration is increased, due to the increased interaction between DMSO molecules and isopropyl groups, DMSO molecules enter the gel, causing the polymer to swell. In summary, solvent-solvent and polymer-solvent competition are responsible for reentrant conformational transitions in the network structure.

The graphs in Figure 12 give inverse bell-shaped curves. By adapting the Gaussian 4 parameter relation [41,42] to determine the transition points (reentrant points, RP) of solvent concentration responsive swelling of hydrogels, the following equation can be written;
(9)$$S=\;\varphi +{\;\omega \;e}^{-0.5\;{\left(\frac{\left(C-\mathrm{RP}\right)}{\sigma }\right)}^{2}}$$

Four shape-controlling parameters, φ, ω, RP and σ, where φ is a base-line swelling value (g g_SH_^−1^), ω is the maximum height (amplitude or the magnitude of the hydrogel’s responsivity, g g_SH_^−1^) that can be achieved on the S-axis, RP is the curve-center (or reentrant points, C_DMSO_%) on the C-axis, and σ is the standard deviation which controls the width of the curve along the C-axis.

The solvent concentration-responsive swelling parameters were found from Equation (9). The correlation coefficients of the graphs drawn according to the proposed equation are 0.986 and above, indicating that the Gaussian 4-parameter equation can be used to find the transition points of smart hydrogels.

Table 8 presents the values of φ, ω, RP, and σ with standard error (SE) found from Equation (6), and RP_S_ (swelling in RP value).

Negative φ values of the hydrogels indicate that swelling decreases with increasing DMSO concentration up to the reentrant transition zone. These φ values, which also indicate the magnitude of the solvent concentration responsibility, are around 3 g g_SH_^−1^ for all three hydrogels. In the swelling curves, the RP value, which gives the minimum swelling value in the shrunken state, is 49.3% for SH-O, while this value is approximately 1% smaller for other hydrogels.

Solvent type-responsive swellings

The hydrogels were immersed in large amounts of solvents (benzene, tetrahydrofuran, acetone, 1-butanol, 1-propanol, ethanol, methanol, and ethanolamine) having increased solubility parameter values at 25 °C until equilibrium was attained. The swelling ratio of the hydrogels (Q = V_s_/V_d_) was calculated by assuming additively of volumes, where V_s_ and V_d_ are the volumes of swollen and dry gel samples, respectively.

To find the solubility parameters of the hydrogels and to examine the solvent-responsible swelling behavior, the solubility parameter (δ) or hydrogen bonding component (δ_H_) values [43,44] of various solvents against the swelling ratio of hydrogels are plotted and given in Figure 13. Due to the hydrophilic nature of the hydrogels, the hydrogen bonding component values of the solubility parameter were used.

The graphs gave bell-shaped curves. By adapting the Gaussian 4 parameter relation [41,42] to determine the solubility parameters [δ_SH_, (MPa)^1/2^] or hydrogen bonding component (δ_H, SH_) of the hydrogels, the following equation can be written;
(10)$$S=\;\psi +{\;\Upsilon e}^{-0.5{\left(\frac{\left(\delta -\delta_{\mathrm{SH}}\right)}{\sigma }\right)}^{2}}$$

Four shape-controlling parameters, ψ, Υ, δ_SH_, and σ, where φ is a base-line swelling value (g g_SH_^−1^), Υ is the maximum height (amplitude or the magnitude of the hydrogel’s responsivity, g g_SH_^−1^) that can be achieved on the Q-axis, δ_SH_ is the curve-center (or solubility parameters [δ_SH_, (MPa)^1/2^]) on the δ-axis, and σ is the standard deviation which controls the width of the curve along the δ-axis. In this equation, when δ_H_ values of the solvents are used instead of δ values, δ_SH, H_ can be used instead of δ_SH_.

The solvent-responsive swelling parameters were found from Equation (10) and shown in Table 9a.

Table 9b exhibits the values of ψ, Υ, δ_SH,_ H and σ with standard error (SE) found from Equation (10).

The correlation coefficients of the graphs drawn according to the proposed equation are 0.995 and above, indicating that the Gaussian 4-parameter equation can be used to find the solubility parameters of smart hydrogels. The solubility parameters of SHs are found to be in the range 26.76–27.74 (MPa)^1/2^.

The hydrogen bonding component values of the solubility parameter of SHs are found to be in the range 20.31–22.293 (MPa)^1/2^. The fact that the hydrogen bonding component values of the solubility parameters of SH-A and SH-M are higher than those of SH-O may indicate that these hydrogels containing acidic groups are capable of making more hydrogen bonds.

### 2.3. Adsorption of Human Serum Albumin

Commercial (CHSA) and blood human serum albumin (BHSA) were used to find the adsorption properties of biopolymers on SHs.

The amount of adsorbed albumin from human blood serum and commercial HSA solution (removal efficiency, RE%) were calculated using the Equation (11);
(11)$$\mathrm{RE}\%=\frac{C_{o}-C_{e}}{C_{o}}\times 100$$
where C_o_ and C_e_ were the initial and equilibrium concentration of human serum albumin in mg L^−1^, respectively [45,46,47,48,49].

The bar graphs of albumin adsorption onto SHs are presented in Figure 14.

In HSA adsorption to smart gels, the adsorption of CHSA is 1.8, 2.3, and 3.1 times higher in SHO, SH-A, and SH-M, respectively, compared to BHSA adsorption. The lower RE% values of BHSA adsorption onto SHs may be due to the complex structure of blood serum.

While the adsorption of BHSA in SH-A and SH-M is 1.4 and 1.8 times higher than that of SH-O, the adsorption of CHSA in hydrogels containing carboxyl group(s) is 1.8 and 3.0 times higher than that of SH-O, respectively.

SH-M was the most efficient SH, adsorbing 49% of CHSA while absorbing 16% of BHSA. The adsorption of BHSA and all the SHs are almost identical, i.e., do not change too much with the type of carboxylic acid monomer in the hydrogel structure. However, hydrogels, which are stimuli-responsive, can be used as chromatographic separations and can be preferred in different formulations e.g., SH-M for CHSA and BHSA. In any case, these types of materials can be used in biomedical fields for various purposes.

## 3. Conclusions

It has been shown here that smart hydrogels responsible for temperature, pH, ionic strength, solvent, and solvent concentration can be easily prepared by thermal free radical redox polymerization and simultaneous crosslinking method using a crosslinker such as NBis. It has been understood that the transition points and responsibility dimension, which are known to be very important in the use of smart hydrogels as biomaterials in the biomedical field, can be easily determined by using Sigmoidal or Gaussian equations. While the LCST value of SH-A was found close to body temperature, it was observed that it could also be used effectively in HSA adsorption.

In conclusion, it can be said that the prepared smart hydrogels such as the SH-O, SH-A, and SH-M can be used as materials in science and technology fields such as biomedicine, biochemistry, biotechnology, immobilization of proteins or biologically active molecules, drug release, environmental, and agriculture. In addition, the use of the proposed sigmoidal equation or Gaussian approaches can be a very useful model for chemists, chemical engineers, biomedicine, polymer, and plastic scientists to determine the transition points of smart hydrogels.

## 4. Materials and Methods

### 4.1. Chemicals

*N*-isopropyl acrylamide (NIPAAm) (Aldrich, Milwaukee, WI, USA), acrylamide (AAm) (Merck Darmstadt, Germany), acrylic acid (A) (Merck Darmstadt, Germany), and maleic acid (M) (Sigma, St. Louis, MO, USA) as monomer or co-monomer, *N*,*N*′-methylenebisacrylamide (NBis) (Merck, Schuchardt, Germany) as crosslinkers, ammonium persulfate (APS) (Merck, Schuchardt, Germany) as redox initiator and *N*,*N*,*N*′,*N*′-tetramethylethylenediamine (TEMED) (Sigma, St. Louis, MO, USA) as catalyst were analytical grade, and were used as received. Double-distilled water was used for all the experiments.

### 4.2. Preparation of Hydrogels

8.1 moles of NIPAAm and 1.0 moles of AAm monomers were dissolved in water, along with 0.9 moles of the dienoic acid comonomer (A or M) containing carboxylic acid group, 0.5 moles of N, 0.1 moles of APS, and 0.1 moles of TEMED were added and filled into plastic pipettes. The hydrogels were labeled SH-A and SH-M, respectively. A copolymer named SH-O was synthesized from the mixture made with the same formulation, which does not contain carboxylic acid. Hydrogels were prepared by thermal free radical solution polymerization at 70 °C in a thermostated water bath. The resulting crosslinked polymer rods were cut into 3–4 mm long pieces and washed with distilled water, dried in air and vacuum, and stored for use in experiments.

### 4.3. Characterization

FTIR spectra were recorded with FTIR Nicolet-520 spectrophotometer in the 4000–400 cm^−1^ range, on grinded hydrogel pelled with KBr, and 30 scans were taken at 4 cm^−1^ resolution.

Thermal analysis was carried out using TG and DSC (Shimadzu-50 model Thermal Analyzer). Thermal analyses were performed employing 10 mg samples in a platinum pan heating up to 550 °C under nitrogen gas flow rate of 25 mL min^−1^ with a heating rate of 10 °C min^−1^.

### 4.4. Swelling

In the swelling studies of smart gels, equilibrium and cyclic equilibrium swellings were performed gravimetrically in different aqueous environments (with stimuli such as temperature, pH, organic solvent, solvent concentration, and ionic strength). For pH-responsive swelling studies, smart hydrogels were swollen at different pHs of the adjusted solutions using HCl and NaOH ranging from 2 to 9 at 25 or 40 °C. For temperature-responsive swelling studies, hydrogels were heated in HCl solution adjusted to pH 3 or NaOH solution adjusted to pH 8.0, with total ionic strength fixed with 0.05 M NaCl, at temperatures in the range of 10–60 °C until swelling equilibrium was reached. The ionic strength-responsive swelling behavior was investigated using various NaCl concentrations in the range of 0.05–1.0 M at 25 °C. Ion/counter ion- responsive swelling studies, the effects of various ions, were performed using NaCl, NaNO_3_, and CaCl_2_ solutions prepared with equal ionic strength. Organic solvent-responsive swelling of the hydrogel was investigated in different organic solvents such as benzene, tetrahydrofuran, acetone, 1-butanol, 1-propanol, ethanol, methanol, ethanolamine, 2-propanol, and dimethylsulfoxide. An incubation time of 24 h at 25 °C was chosen. For temperature and pH-responsive cyclic equilibrium swelling of smart hydrogels, temperatures of 25 °C (below the LCST value) and 40 °C (above the LCST) and pH 3 (below the inflection point value) and pH 8 (above the inflection point value) were selected. It was carried out alternately with swelling up to the maximum swelling values for 8 days, first at 25 °C for 24 h at pH 3 and then for 24 h at pH 8. The same process was repeated at 40 °C (above LCST). In all swelling experiments, the increase in mass was calculated by subtracting the swollen hydrogel from the fluid medium and weighing it.

### 4.5. Calculations

Model fittings were made using the SigmaPlot Version 12.5 (Systat Software Inc., Slough, UK) software and the parameters were automatically obtained with standard error (SE) and correlation coefficients (r).

### 4.6. Adsorption of Human Serum Albumin

Commercial human serum albumin (CHSA) and human blood serum albumin (BHSA) were used in the adsorption experiments of albumin in hydrogels. 10 mg L^−1^ HSA solutions were prepared in a universal buffer solution (pH = 3). 0.1 g of dry SHs was put into these solutions and incubated in a water bath at 40 °C for 24 h.

## Figures and Tables

**Figure 1 gels-07-00113-f001:**
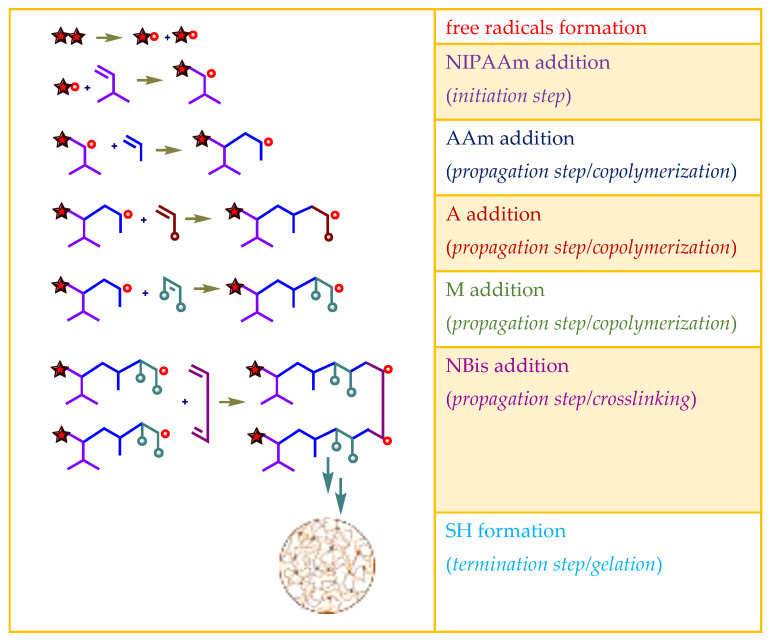
The plausible polymerization and crosslinking mechanism of the hydrogels.

**Figure 2 gels-07-00113-f002:**
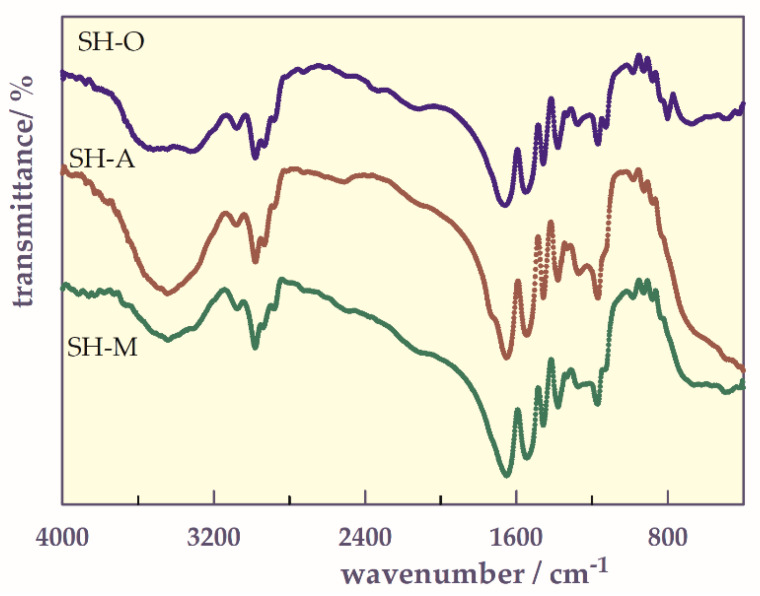
FTIR spectra of the hydrogels.

**Figure 3 gels-07-00113-f003:**
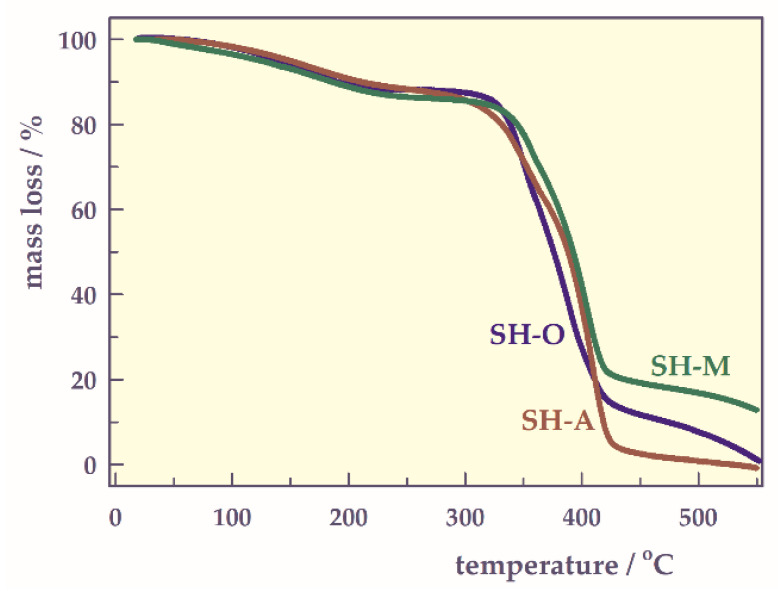
TG thermograms of the hydrogels.

**Figure 4 gels-07-00113-f004:**
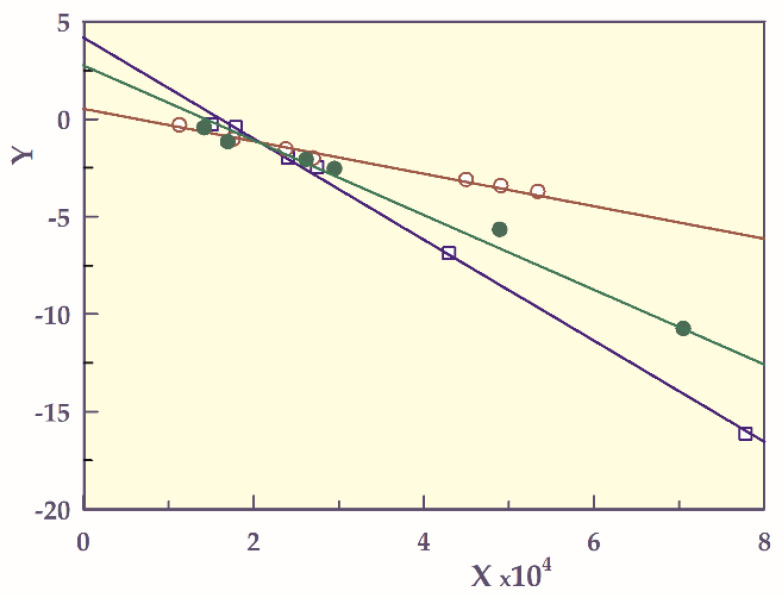
Freeman-Carroll graphs of SHs, **□**; SH-O, o; SH-A, ●; SH-A, and ──; model fit.

**Figure 5 gels-07-00113-f005:**
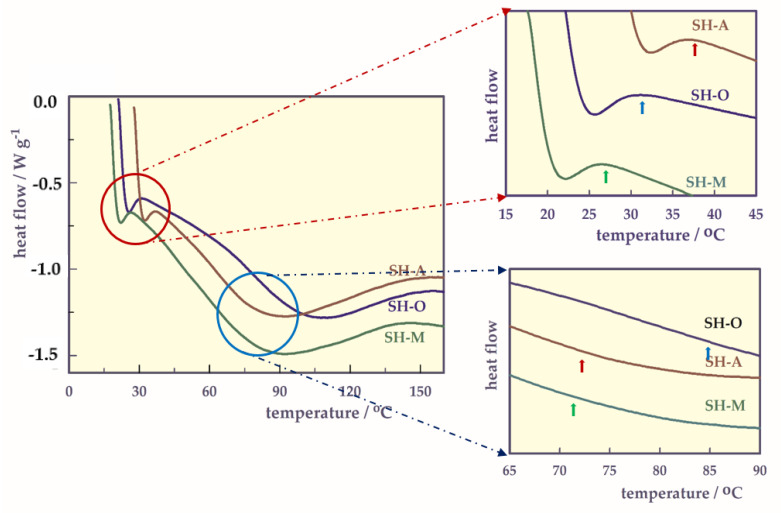
DSC thermograms of the hydrogels.

**Figure 6 gels-07-00113-f006:**
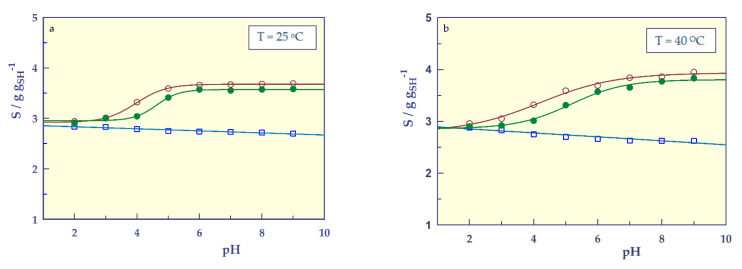
pH-responsive swelling of the SHs at 25 ^O^C (**a**) and at 40 ^O^C (**b**), **□**; SH-O, o; SH-A, ●; SH-M, and ──; model fit.

**Figure 7 gels-07-00113-f007:**
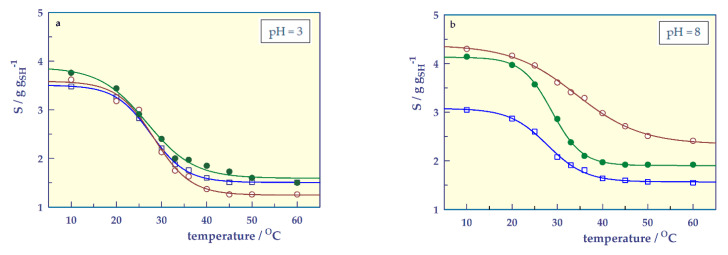
Temperature-responsive swelling of the SHs at pH = 3 (**a**) and at pH = 8 (**b**), **□**; SH-O, o; SH-A, ●; SH-M, and ──; model fit.

**Figure 8 gels-07-00113-f008:**
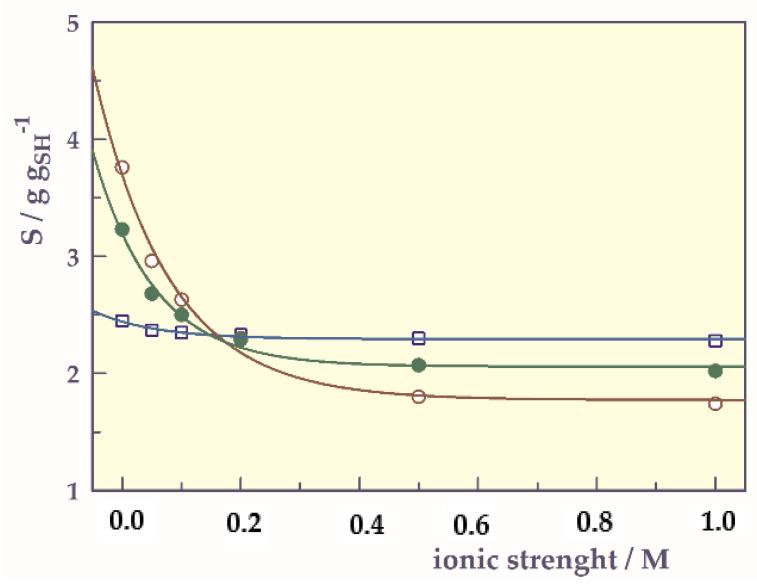
Ionic strength-responsive swelling of SHs, **□**; SH-O, o; SH-A, ●; SH-M, and ──; model fit.

**Figure 9 gels-07-00113-f009:**
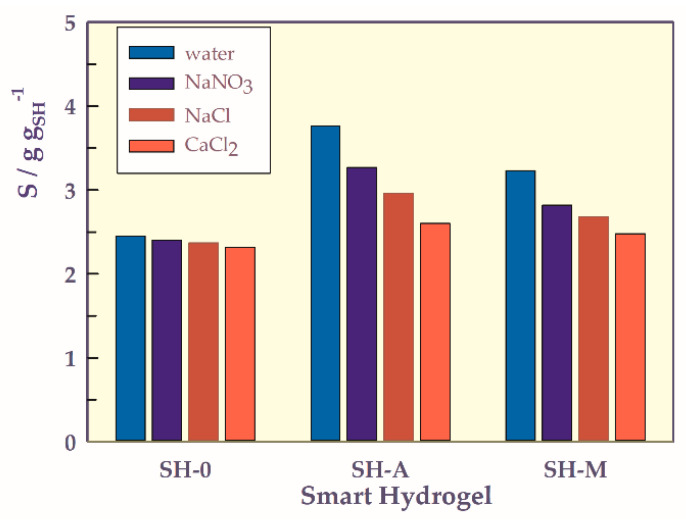
Ion and counter ion-dependent swelling bar graph of SHs.

**Figure 10 gels-07-00113-f010:**
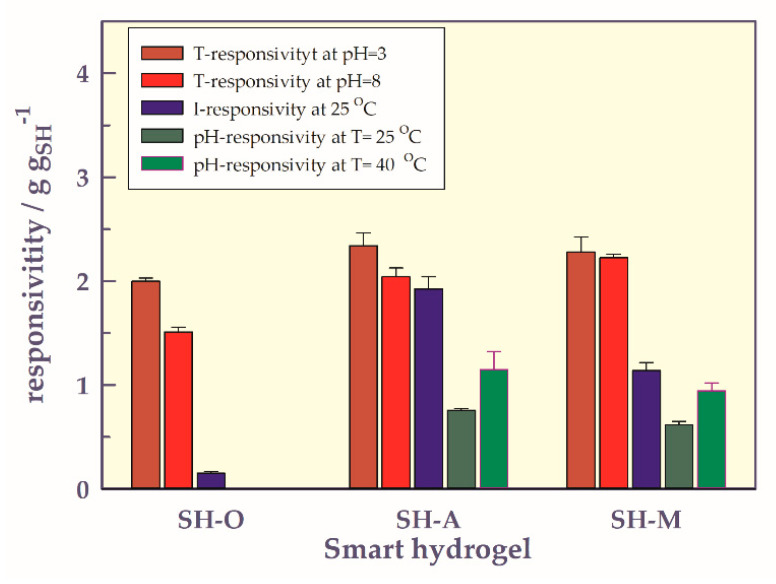
Comparison of stimuli responsivities.

**Figure 11 gels-07-00113-f011:**
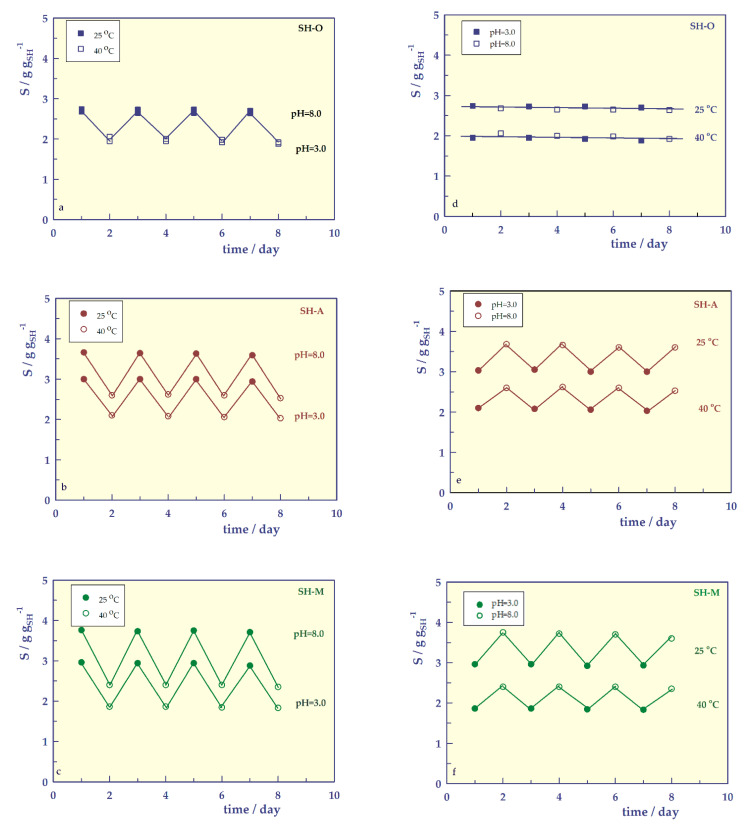
Temperature-responsive reversible swelling profiles of SHs at pH = 3 and at pH = 8; (**a**) SH-O, (**b**) SH-A, (**c**) SH-M, and pH-responsive reversible swelling profiles of SHs at 25 °C and at 25°; (**d**) SH-O, (**e**) SH-A, (**f**) SH-M.

**Figure 12 gels-07-00113-f012:**
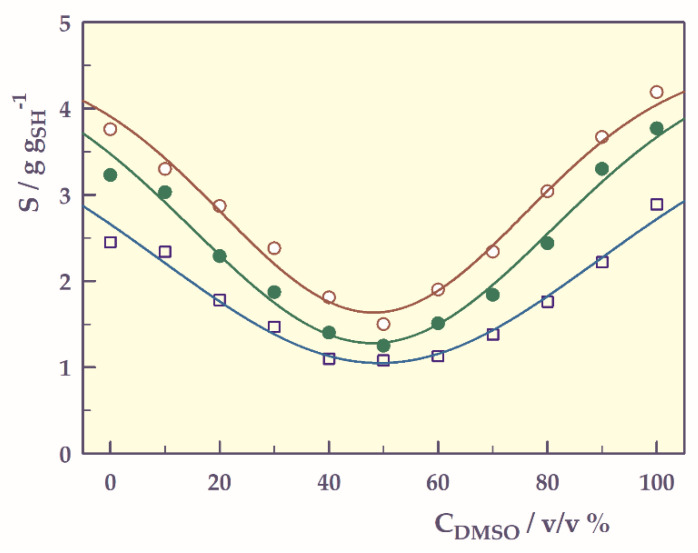
Solvent concentration-responsive swelling of the SHs, **□**; SH-O, o; SH-A, ●; SH-M, and ──; model fit.

**Figure 13 gels-07-00113-f013:**
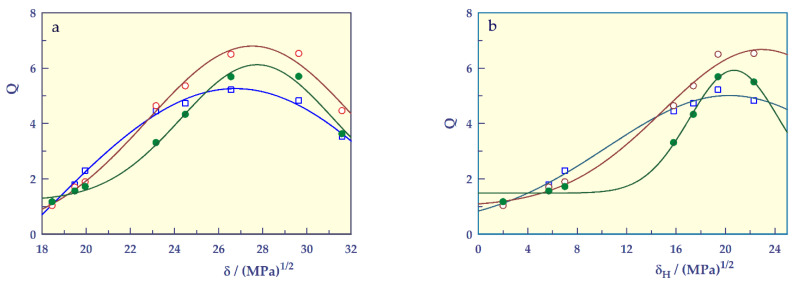
Solvent-responsive swelling of the SHs, (**a**) solubility parameter and (**b**) hydrogen bonding component, **□**; SH-O, o; SH-A, ●; SH-M, and ──; model fit.

**Figure 14 gels-07-00113-f014:**
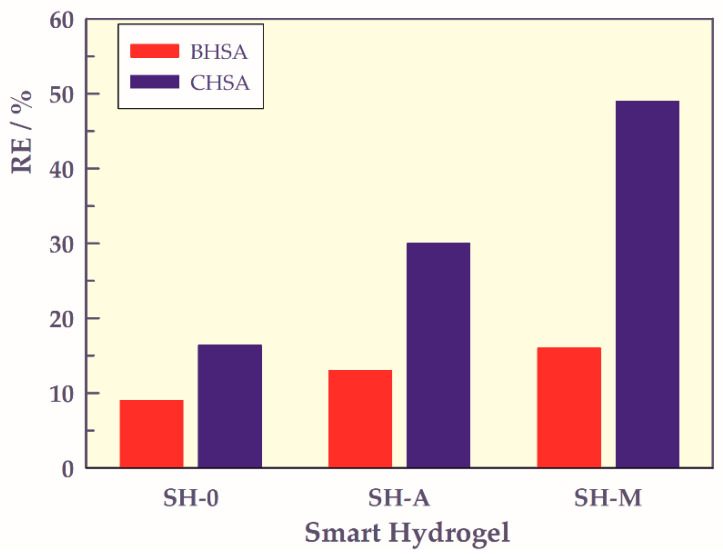
The bar graph of human serum albumin adsorption onto SHs.

**Table 1 gels-07-00113-t001:** The chemical structure and some properties of the monomers.

Monomer	Abbreviation	Chemical Structure	Illustration	Molar Mass/g mol^−1^
*N*-isopropylacrylamide *(2-propenamide N-(1-methylethyl)-)*	NIPAAm	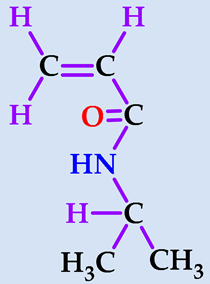	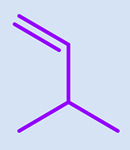	113.16
acrylamide *(**2-propenamide)*	AAm	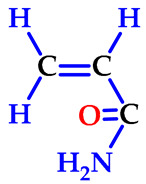	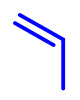	71.08
acrylic acid (*propenoic acid*)	AA	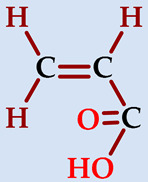	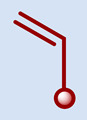	72.06
maleic acid (*cis-butenedioic acid*)	MA	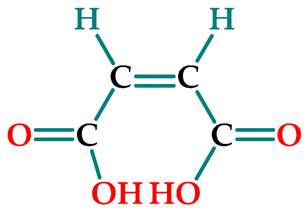	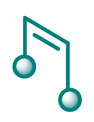	116.06
*N*,*N*′-methylenebisacrylamide	NBis	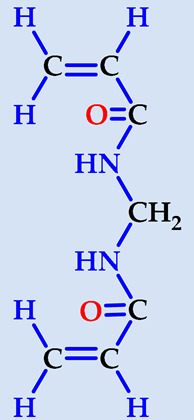	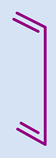	154.17

**Table 2 gels-07-00113-t002:** Thermogravimetric parameters of the hydrogels.

SH	T_i_/°C	T_max_/°C	T_f_/°C	T_h_/°C	R_max_/mg min^−1^	C_max_/%
SH-O	330	382	415	374	0.85	44
SH-A	335	400	436	390	0.85	41
SH-M	340	396	427	390	0.87	45

**Table 3 gels-07-00113-t003:** Kinetic parameters of thermal degradation of SHs.

Parameters	SH-0	SH-A	SH-M
−E_FC_/R ± SE	25,911.217 ± 712.623	8325.565 ± 207.644	19,191.567 ± 718.121
*n*_FC_ ± SE	4.181 ± 0.288	0.529 ± 0.110	2.752 ± 0.404
E_FC_	215.43	69.287	159.392
r	0.999	0.998	0.997
*n* _J_	0.049	0.022	0.044
E_J_	115.159	59.343	109.974

**Table 4 gels-07-00113-t004:** LCST and glass transition temperature (T_g_) of the hydrogels.

SH	LCST/°C	T_g_/°C
SH-O	31.24	84.82
SH-A	36.58	72.20
SH-M	27.41	71.88

**Table 5 gels-07-00113-t005:** The parameters of pH-responsive swelling at I = 0.05 M.

T	25 °C			40 °C		
Parameter	SH-O	SH-A	SH-M	SH-O	SH-A	SH-M
α ± SE	-	0.757 ± 0.016	0.617 ± 0.034	-	1.149 ± 0.172	0.945 ± 0.073
β ± SE	-	0.484 ± 0.034	0.356 ± 0.080	-	1.177 ± 0.284	0.838 ± 0.165
IP ± SE	-	3.964 ± 0.036	4.626 ± 0.112	-	4.198 ± 0.355	5.170 ± 0.168
S_o_ ± SE	-	2.920 ± 0.013	2.954 ± 0.027	-	2.785 ± 0.141	2.859 ± 0.049
r	-	0.999	0.996	-	0.996	0.996
S_max_	-	3.571	3.677	-	3.934	3.804
IP_S_	-	3.262	3.080	-	3.360	3.332
S_i (at pH = 3)_	2.70	2.90	2.94	2.62	2.96	2.90
S_f (at pH = 9)_	2.84	3.58	3.69	2.88	3.95	3.83

**Table 6 gels-07-00113-t006:** The parameters of temperature-responsive swelling at pH = 3 and pH = 8.

pH	3			8		
Parameter	SH-0	SH-A	SH-M	SH-0	SH-A	SH-M
ϕ ± SE	1.999 ± 0.031	2.342 ± 0.122	2.281 ± 0.145	1.510 ± 0.047	2.042 ± 0.083	2.226 ± 0.031
Ψ ± SE	–3.929 ± 0.152	–4.077 ± 0.535	–5.192 ± 0.707	–4.219 ± 0.319	–7.152 ± 0.576	–3.321 ± 0.133
LCST ± SE	27.601 ± 0.191	28.303 ± 0.653	26.745 ± 0.853	27.835 ± 0.392	34.215 ± 0.535	28.798 ± 0.164
S_o_ ± SE	1.508 ± 0.014	1.247 ± 0.056	1.593 ± 0.057	1.567 ± 0.021	2.340 ± 0.047	1.904 ± 0.015
r	1.000	0.996	0.996	0.999	0.999	1.000
S_max_	3.507	3.589	3.874	3.077	4.382	4.130
LCST_S_	2.508	2.419	2.730	2.321	3.362	3.017
S_i (at T = 10 °__C)_	1.51	1.26	1.50	1.55	2.41	1.92
S_f (at T = 60 °__C)_	3.48	3.62	3.76	3.05	4.30	4.14

**Table 7 gels-07-00113-t007:** The parameters of ionic strength -responsive swelling.

Parameters	SH-O	SH-A	SH-M
S_o_ ± SE	2.293 ± 0.011	1.774 ± 0.081	2.059 ± 0.048
ξ ± SE	0.150 ± 0.018	1.923 ± 0.121	1.138 ± 0.077
λ ± SE	9.714 ± 2.677	7.806 ± 1.186	9.723 ± 1.525
r	0.980	0.994	0.993
S_max_ (S_o_ + ξ)	2.443	3.196	3.696
S_i (at I = 0 M)_	2.28	2.02	1.74
S_f (at I = 1 M)_	2.45	3.23	3.76

**Table 8 gels-07-00113-t008:** The parameters of solvent concentration-responsive swelling.

Parameters	SH-O	SH-A	SH-M
φ ± SE	4.052 ± 1.264	4.550 ± 0.341	4.609 ± 0.674
ω ± SE	−3.002 ± 1.225	−2.916 ± 0.312	−3.328 ± 0.644
RP ± SE	49.300 ± 1.041	48.174 ± 0.831	47.983 ± 0.903
σ ± SE	39.771 ± 12.151	27.679 ± 3.281	32.693 ± 5.650
r	0.986	0.992	0.990
RP_S_	1.051	1.635	1.281

**Table 9 gels-07-00113-t009:** (**a**) The solvent-responsive swelling parameters of hydrogels for δ. (**b**) The solvent-responsive swelling parameters of hydrogels for δ_H_.

**(a)**			
**Parameters**	**SH-O**	**SH-A**	**SH-M**
ψ ± SE	–6.503 ± 7.271	–0.005 ± 0.808	1.208 ± 0.201
Υ ± SE	11.765 ± 7.205	6.797 ± 0.700	4.914 ± 0.257
δ_SH_ ± SE	26.757 ± 0.140	27.524 ± 0.176	27.742 ± 0.135
σ ± SE	8.849 ± 3.316	4.742 ± 0.637	3.441 ± 0.257
r	0.997	0.995	0.996
**(b)**			
**Parameters**	**SH-O**	**SH-A**	**SH-M**
ψ ± SE	0.254 ± 0.744	0.998 ± 0.489	1.484 ± 0.138
Υ ± SE	4.758 ± 0.733	5.676 ± 0.682	4.439 ± 0.248
δ_SH,H_ ± SE	20.307 ± 0.962	22.937 ± 1.941	20.703 ± 0.324
σ ± SE	9.914 ± 1.892	8.036 ± 2.149	3.620 ± 0.415
r	0.997	0.996	0.996

## Data Availability

Not applicable.

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
