# Peer review of "Smart Hydrogels: Preparation, Characterization, and Determination of Transition Points of Crosslinked N-Isopropyl Acrylamide/Acrylamide/Carboxylic Acids Polymers"

_gels, 2021, doi:10.3390/gels7030113_

Round 1

Reviewer 1 Report

Işıkver et al. report the copolymerization of smart thermoresponsive hydrogels of NIPAAm with AAM in which acidic comonomers as Acrylic acid and Maleic acid are incorporated. Authors confirmed that the obtained hydrogel respond to both temperature and pH. Besides of preparation and characterization by means of FTIR, TGA, DSC, swelling behavior in response to pH variation, temperature, ionic strength and solvent type are also determined. Although much research has been already performed regarding stimuli responsive NIPAAm-AAM hydrogels with acidic comonomers, the approach used in this manuscript lead to interesting parameters such as the responsibility dimensions making possible to compare different hydrogel compositions with their responsibility to different stimuli, which would allow to choose at a glance the most suitable hydrogel for the required application. Overall, the work can be considered for the journal if some revisions have been conducted.

Revisions:

In general, I will strongly recommend to improve the quality of all of the figures. As they are right now it is difficult to read the results explained and discussed in the text.

Page 5, line 128: When analyzing the results of TGA teste authors indicated the following: “Accordingly, the incorporation of the M-containing dicarboxylic unit into SH-M gives flexibility to the polymeric structure, while the incorporation of the A-containing monocarboxylic group into SH-A makes it more rigid”. The terms flexibility and rigidity do not correspond to a TGA analysis. This characterization should be discussed in terms of thermal stability, inducing a more or less stable hydrogel. This analysis need to be reevaluated.

Page 6, line 156-160: Is there any advantage of having a more easily degradable hydrogel, as is the case for SH-A? As it is written it seem that having a more easily degradable hydrogel is relevant to the work. A sentence or two clarifying this point is needed.

Page 8 , line 179: The Tg collected in table 4 are not evident in the thermogram. ¿Could the author indicate where are these transitions in each hydrogel? This point it is not clear.

Page 11, line 275: The equation 5 is divided in 3 lines. This should be corrected.

Page 11, line 280: This sentence need to be rewritten it is difficult to understand: “The correlation coefficients of the graphs drawn according to the proposed 280 equation are in the range of 0.98 to 0.99, indicating that the equation with exponential 281 decay single, 3 parameters can be used to determine the ionic strength responsive swelling 282 parameters of smart hydrogels.”

Page 11, line 292: The sentence need English correction: “Comparing the effect of ion and counter ion for SH swelling ratios at equal ionic strength (0.05 M was chosen, where all hydrogels are responsible) are shown in Figure 9.”

Page 12, line 311: Sentence to be revised. “…..prepared hydrogels against stimuli such as temperature, pH, and ionic strength, and to be more understandable.”

Page 12, line 326: Author stated that acrylamide provide mechanical strength to the hydrogels, however, hydrogels mechanical properties have not been characterized, nor rheological properties. In order to confirm such statement, rheological measurements of hydrogels should be performed.

Author Response

Reviewer 1

First of all, we would like to express our sincere thanks and respect for your comments and suggestions that took your valuable time. Your suggestions, our responses to them, and corrections are as presented below. We have also added the corrections made in line with your suggestions to the manuscript.

In general, I will strongly recommend to improve the quality of all of the figures. As they are right now it is difficult to read the results explained and discussed in the text. 

Page 5, line 128: When analyzing the results of TGA teste authors indicated the following: “Accordingly, the incorporation of the M-containing dicarboxylic unit into SH-M gives flexibility to the polymeric structure, while the incorporation of the A-containing monocarboxylic group into SH-A makes it more rigid”. The terms flexibility and rigidity do not correspond to a TGA analysis. This characterization should be discussed in terms of thermal stability, inducing a more or less stable hydrogel. This analysis need to be reevaluated.

Yes, you are right. The sentence: “Accordingly, the incorporation of the M-containing dicarboxylic unit into SH-M gives flexibility to the polymeric structure, while the incorporation of the A-containing monocarboxylic group into SH-A makes it more rigid.” has been removed and replaced with the following sentence; “These values indicate that SH-M containing two carboxyl groups in its structural repeating unit degrades at slightly lower temperatures than SH-A containing a single carboxyl group. The presence of more than one carboxyl group in the structural repeating unit of SH slightly facilitated thermal degradation.”

Page 6, line 156-160: Is there any advantage of having a more easily degradable hydrogel, as is the case for SH-A? As it is written it seem that having a more easily degradable hydrogel is relevant to the work. A sentence or two clarifying this point is needed.

Here is a comparison between hydrogels and as a result of the calculated parameters, it is stated that the thermal degradation of one polymer is easier than the other. Considering that these hydrogels can be used at lower temperatures than the thermal decomposition temperatures (such as room, outdoor or body temperatures), there is no advantage that thermal decomposition is easy. Since these evaluations were made in terms of characterization of hydrogels, it was preferred not to write a new additional sentence.

Page 8, line 179: The Tg collected in table 4 are not evident in the thermogram. ¿Could the author indicate where are these transitions in each hydrogel? This point it is not clear.

Yes, the glass transition temperature values in the DSC thermograms cannot be seen very clearly. However, the points in the DSC thermogram are given with a software as numerical data with very small intervals. Tg values can be found by analyzing the numerical data of these thermograms in the inflection regions.

Page 11, line 275: The equation 5 is divided in 3 lines. This should be corrected.

Sorry. Although Equation 5 is written in a single line, it appears as 3 lines in PDF format for some reason. Equation 5 has been corrected.

Page 11, line 280: This sentence need to be rewritten it is difficult to understand: “The correlation coefficients of the graphs drawn according to the proposed 280 equation are in the range of 0.98 to 0.99, indicating that the equation with exponential 281 decay single, 3 parameters can be used to determine the ionic strength responsive swelling 282 parameters of smart hydrogels.”

Yes, you are right. The sentence: “The correlation coefficients of the graphs drawn according to the proposed equation are in the range of 0.98 to 0.99, indicating that the equation with exponential decay single, 3 parameters can be used to determine the ionic strength responsive swelling parameters of smart hydrogels.” has been removed and replaced with the following sentence; “The correlation coefficients (r) of the graphs drawn according to the proposed equation were found in the range of 0.98 to 0.99. The r values close to the unity indicate that the exponential decay single, 3 parameters equation can be used to determine the ionic strength responsive swelling parameters of smart hydrogels.”

Page 11, line 292: The sentence need English correction: “Comparing the effect of ion and counter ion for SH swelling ratios at equal ionic strength (0.05 M was chosen, where all hydrogels are responsible) are shown in Figure 9.”

The sentence: “Comparing the effect of ion and counter ion for SH swelling ratios at equal ionic strength (0.05 M was chosen, where all hydrogels are responsible) are shown in Figure 9.” has been deleted and replaced with the following sentence; To compare the ion and counter ion influences on the swelling of SHs in solutions of NaNO3, NaCl and CaCl2 with an ionic strength of 0.05 M, and in water, a bar graph is plotted and shown in Figure 9.

Page 12, line 311: Sentence to be revised. “…..prepared hydrogels against stimuli such as temperature, pH, and ionic strength, and to be more understandable.”

The sentence: “Bar graphs were prepared and presented in Figure 10 to compare the responsibilities of the prepared hydrogels against stimuli such as temperature, pH, and ionic strength, and to be more understandable.” have been revised as “To compare the responsibilities of SHs to stimuli such as temperature, pH, ionic strength, a bar chart was prepared and presented in Figure 10.”

Page 12, line 326: Author stated that acrylamide provide mechanical strength to the hydrogels, however, hydrogels mechanical properties have not been characterized, nor rheological properties. In order to confirm such statement, rheological measurements of hydrogels should be performed.

NiPAAm/carboxylic acid hydrogels prepared without the addition of AAm broke into small pieces when kept in water for a certain time. To prevent this fragmentation, AAm was added to the hydrogels during preparation. The term "mechanical strength" is used because the hydrogels do not break down as a result of this process. This very ambitious term has been replaced by the word “durability enhancement property”.

Reviewer 2 Report

why is absorption of albumin of interest?

Author Response

Reviewer 2

First of all, we would like to express our sincere thanks and respect for your comments and suggestions that took your valuable time. Your suggestion and our response and correction are as presented below.

why is absorption of albumin of interest?

With the idea that the prepared smart hydrogels could be a potential biomaterial, the adsorption of human serum albumin in the gel was also examined. Human sera-gel interaction was also presented in a previously published study (Reference 27).

Reviewer 3 Report

In the paper, the authors prepared the acrylamide hydrogel with NIPAAm, acrylic acid and maleic acid to investigate the LSCT and swelling properties of hydrogel with different densities of the carboxylic acid group in the polymer. The results also fitted by empirical using sigmoidal equations.  Some concerns must be addressed before publication.

  1. The smart hydrogel was prepared with thermal initiated free radical polymerization. Did the authors characterize the conversion rate of the monomers?
  2. Authors calculated IP, LSCT using sigmoidal equations from experimental data. Could authors compare the calculated values with experimental values? For example, LSCT acquired by DSC, IP by swelling test and the calculated value.
  3. The details of sigmoidal equation fitting is missing in the methods. How is SE of the fitted parameter calculated in Tables 5 to 8? How many samples were tested for repeatability?
  4. Equation 7 may not be correct. The pH in the equation should be Temperature.
  5. The current title is not helpful to the readers. It looks like a review article. Please modified the title to include more details.
  6. The figure quality of the manuscript must be improved. Please do not use screen capture to get the images.

Author Response

Reviewer 3

First of all, we would like to express our sincere thanks and respect for your comments and suggestions that took your valuable time. Your suggestions, our responses to them, and corrections are as presented below. We have also added the corrections made in line with your suggestions to the manuscript.

In the paper, the authors prepared the acrylamide hydrogel with NIPAAm, acrylic acid and maleic acid to investigate the LSCT and swelling properties of hydrogel with different densities of the carboxylic acid group in the polymer. The results also fitted by empirical using sigmoidal equations.  Some concerns must be addressed before publication.

  1. The smart hydrogel was prepared with thermal initiated free radical polymerization. Did the authors characterize the conversion rate of the monomers?

The conversion rate of monomers has not been characterized.

  1. Authors calculated IP, LSCT using sigmoidal equations from experimental data. Could authors compare the calculated values with experimental values? For example, LSCT acquired by DSC, IP by swelling test and the calculated value.

The difference in LSCT values found from DSC and swelling values in lines 254 to 256 on page 10 was mentioned, and it was stated that the difference between the two methods might be due to the swelling environment.

  1. The details of sigmoidal equation fitting is missing in the methods. How is SE of the fitted parameter calculated in Tables 5 to 8? How many samples were tested for repeatability?

All model fittings given in the article were made with the use of SigmaPlot Version 12.5 (Systat Software Inc. UK) software and the standard error (SE) and correlation coefficients (r) of the found parameters were obtained automatically. In addition, the following sentence has been added to the Materials and Methods section. "4.5. Calculations Model fittings were made with the use of SigmaPlot Version 12.5 (Systat Software Inc. UK) software and parameters were obtained automatically with standard error and correlation coefficients."

As for the repeatability tests, as given in the reversible swelling part, consecutive swelling tests were carried out for 8 days, and it was observed that the swelling values in the same environment (for example, at the same temperature and pH) were at the same values. Considering this situation, it was not necessary to carry out repeatability tests.

  1. Equation 7 may not be correct. The pH in the equation should be Temperature.

Yes, I'm sorry. pH notation in equation 7 has been replaced by T notation.

  1. The current title is not helpful to the readers. It looks like a review article. Please modified the title to include more details.

Thanks. Upon your suggestion, the title has been corrected as follows to be more descriptive. "Smart Hydrogels: Preparation, Characterization, and determination of transition points of crosslinked N-isopropyl acrylamide/acrylamide/carboxylic acids polymers”

  1. The figure quality of the manuscript must be improved. Please do not use screen capture to get the images.

All figures in the text were prepared at 300 dpi resolution and added to the manuscript. I guess when the shapes are converted to PDF format, they look like screenshots. All figures were re-examined at their resolutions, they were found to be 300 DPI, and the texts in the figures were bolded and put back in the manuscript.

Round 2

Reviewer 3 Report

The authors addressed my comments.